# A Methodology to Analyze the Presence of Sustainability in Engineering Curricula. Case of Study: Ten Spanish Engineering Degree Curricula

**Fermín Sánchez-Carracedo [1],\* , Francisco Manuel Moreno-Pino [2], Bárbara Sureda [1] , Miguel Antúnez [3] and Ibon Gutiérrez [4]**

[1]  University Research Institute for Sustainability Science and Technology (IS.UPC), Universitat Politècnica de Catalunya–BarcelonaTech, Campus Nord, C/Jordi Girona 34, 08034 Barcelona, Spain
[2]  Departmento de Didáctica, University of Cádiz, 11519 Cádiz, Spain
[3]  Servicio de Protección Ambiental, University of Córdoba, 14014 Córdoba, Spain
[4]  Facultad de Formación de Profesorado y Educación, Universidad Autónoma de Madrid, 28049 Madrid, Spain
\*  Correspondence: fermin@ac.upc.edu

**Abstract:** This paper presents a methodology to analyze the sustainability presence level in the curriculum of an engineering degree. The methodology is applied to ten engineering degrees of the Spanish university system, taught in three different universities. The design used for the research is quantitative and correlational. The analytical instrument used is the engineering sustainability map, which contains the learning outcomes related to sustainability that are expected of engineering students upon completion of their studies. The methodology is used to analyze the curricula of the ten engineering degrees in order to identify what learning outcomes of the engineering sustainability map are developed in each degree. The results indicate that the sustainability competency least present in all the degrees is the "participation in community processes that promotes sustainability," with an average presence of 23.3%, while the most present is the "application of ethical principles related to the values of sustainability in personal and professional behavior," with an average presence of 76.6%. In general, learning outcomes related to sustainability have an average presence of 52.1%, so practically half of the cells in the ten engineering sustainability maps are not developed in the degrees under study.

**Keywords:** engineering education; sustainability in engineering degrees; sustainability competencies; engineering sustainability map; sustainability presence map; education for sustainable development; curriculum design; learning outcomes

## 1. Introduction

### 1.1. Motivation and Related Work

The world we live in is characterized by its enormous complexity and uncertainty. It is a complex interconnected system that presents multiple problems such as poverty, hunger, gender inequality, access to education, climate change, deforestation, and loss of biodiversity. New approaches are required to analyze and solve all these problems [1].

Over recent decades, a new field of knowledge has emerged: Sustainability science, which seeks to understand the fundamental nature of the interactions between nature and society [2]. Sustainability science must be profoundly interdisciplinary and transdisciplinary in order to be able to tackle complex challenges. In addition, it must have a broad perspective, both spatial and temporal.

Furthermore, for the last few decades, the United Nations has attempted to implement strategies to achieve sustainable development (although there is no unanimity on whether the terms sustainability and sustainable development refer to the same concept, both terms will be used as synonyms in this paper for the sake of simplicity) that facilitates the resolution of some of the aforementioned problems. These strategies include the approval of the Millennium Development Goals in 2000, which were to be achieved in 2015 [3]. The Millennium Development Goals were a set of eight objectives that have served as a stimulus to meet the needs of the poorest, although the scope of each objective was uneven. The 2030 Agenda for Sustainable Development continues the work begun with the Millennium Development Goals. The 2030 agenda contains a set of 17 objectives, known as Sustainable Development Goals, which are to be achieved by the year 2030 [4]. The Sustainable Development Goals aim to protect the planet and guarantee peace and prosperity for all the people who inhabit it. Among the Sustainable Development Goals, Objective 4 is highly relevant and aims to guarantee an inclusive, equitable and quality education as well as promoting lifelong learning opportunities for all. The United Nations believes that education is a basic factor that should be taken into account to ensure the sustainable development of our planet and achieve the Sustainable Development Goals by 2030.

Goal 4.7 of Sustainable Development Goal 4 specifically addresses education for sustainable development and recognizes it as a key factor in achieving the sustainability of the planet. Education plays a key synergistic role for the achievement of the aspirations of the 2030 Agenda [5,6]. The Global Action Program on Education for Sustainable Development seeks to accelerate progress towards sustainable development, thus contributing to the 2030 agenda [7]. Education for Sustainable Development seeks to achieve an informed and involved citizenship that develops creative skills that: (1) Allow one to solve problems, (2) provide one with scientific, technological and social literacy, and (3) achieve a commitment to participation in responsible actions that help guarantee an adequate environment and a socially just and economically prosperous future for all [8]. Education, therefore, needs to be reoriented towards a combination of interdisciplinarity, vision, creativity and fun [9].

As an open space for thought, reflection and action, universities must assume a leadership role in the development of strategies and methodologies capable of solving the multiple challenges that arise, both globally and locally. Universities must contribute to the training of active graduates committed to sustainable development, acting as a catalyst for society towards planetary sustainability [10].

Many universities have signed international declarations committing them to introducing sustainable development into their curricula, research and social projection [11]. Curricular sustainability involves the creation of spaces for reflection and collective collaboration, both inter- and trans-disciplinary, that foster learning, critical reflection, and creative and innovative action.

A multitude of subjects focused on Education for Sustainable Development have been implemented in higher education curricula over the last few years. However, when the curricula of the universities that teach engineering studies have been analyzed, it has been observed that the design of the curricula has, as its basic objective, that engineers be able to solve technological problems without considering, in many cases, the social and environmental impact of their work [12].

Engineering education has a long tradition of dealing with environmental issues. The treatment of effluent, municipal solid waste disposal, and energy efficiency [13] has long been a common practice in various engineering schools. However, these problems are usually addressed from the technological perspective of the equipment required instead of in terms of the ecosystem.

The rise of environmental awareness in the 1970s affected engineering schools [14], leading to the inclusion of environmental and energy issues in some curricula, especially civil engineering, architecture and chemical engineering. However, environmental issues have only marginally affected most of the other engineering curricula.

A second wave of environmental awareness was triggered by the publication of the so-called Brundtland report [15]. This renewed interest in environmental issues resulted in new initiatives, but sustainability ethics still needs to form part of institutional culture [16].

The approaches adopted for the introduction of sustainability into engineering education in the 1990s were somewhat naïve; measures such as developing an add-on course, teaching other teachers about sustainable development and creating a track for sustainable development specialists constituted, at best, just an initial step. The next steps to be taken in Education for Sustainable Development should not only address what course should be added to make engineering more sustainable but also the question of what type of curriculum might contribute effectively to sustainable development. Moreover, students and faculty should be motivated to improve their sustainability competencies. Curricula should be rebuilt by benefitting from Education for Sustainable Development expertise as the leading principle for curricula instead of adapting current unsustainable curricula to introduce sustainable development

Engineering students must learn to think long-term and understand that in order to achieve a better world, they must situate their future professional activities within the framework of sustainable solutions. To this end, future engineers must be aware of the complexities of the social environment in which they are developing their work and of the need to harmonize short-term improvements with sustainable development based on the long term [14]. The vision of engineering students seems to be very anthropocentric [17]. The work of engineers in industrial development, water and wastewater management, transportation networks, etc., has a significant impact on daily life, human health, and the environment. This highlights the need to educate engineers from a sustainable perspective [18]. The acquisition of knowledge in areas as diverse as, for example, renewable energies or electronic waste, can be used as a gateway to maintaining and expanding the interest of students and their commitment to other critical challenges of engineering and sustainability [19]. For all these reasons, Education for Sustainable Development in engineering is essential for the training of agents of change and transformation that can promote policies, strategies, and methods that enable a more sustainable future to be built [14].

When the European Higher Education Area was created, Education for Sustainable Development was integrated into the degree programs following the methodology of the Tuning project, according to which the level of training must be achieved in terms of competencies and learning outcomes [20]. The Tuning project considers five competencies related to sustainability [21]:

- C6: Ability to show awareness of equal opportunities and gender issues.
- C17: Ability to act on the basis of ethical reasoning.
- C23: Ability to act with social responsibility and civic awareness.
- C25: Appreciation of and respect for diversity and multiculturalism.
- C28: Commitment to the conservation of the environment.

The competency-based education model has directed the teaching-learning process in university students. Teachers assume the role of tutor/guide/facilitator, while skills and methodologies conducive for education in sustainable development are promoted. The European Higher Education Area therefore provides an opportunity to introduce Education for Sustainable Development in higher education. To achieve this goal, university teaching staff must be trained and motivated to include sustainability in the curriculum [22].

The literature review reflects a wide range of perspectives on what criteria or issues related to sustainable development should be incorporated into the engineering curricula, as well as the terminology to be used [23]. The introduction of sustainability in the degrees has been carried out in three ways:

- Introducing specific sustainability subjects in the curriculum [24].
- Including objectives related to sustainability in technical subjects [25,26].
- Applying criteria of sustainability in the final degree thesis [27,28].

Some studies indicate that this integration has been carried out in a very uneven manner. Depending on the university analyzed, imbalances have been observed when developing the different dimensions of sustainability [29]. In general, universities develop economic and environmental aspects more than social and ethical ones, and a holistic vision of sustainability is not taught. To help students gain a holistic view of sustainable development, they must adopt different approaches [30]. Theoretical classes are not enough, so it is important to help them contextualize theoretical knowledge in professional and daily activities [31].

In conclusion, engineering higher education is not currently developing the sustainability competencies that professionals should and can develop in the industry [32]. An effective methodology to introduce these competencies is to use maps of (competencies in) sustainability [25]. A sustainability map helps to define the way in which sustainability should be developed in a degree and to evaluate the development and acquisition of sustainability in an effective way. The sustainability map contributes to creating awareness about the course content and teaching practices [26]. This map contains the learning outcomes related to sustainability that are expected of graduates at the end of their studies. These learning outcomes must be distributed among a set of subjects that form the sustainability itinerary of the degree. Subjects must design learning activities that allow students to achieve the learning outcomes that have been assigned to them. A sustainability map is, therefore, an essential tool to define how sustainability should be introduced in engineering degrees.

Spanish universities have begun the introduction of Education for Sustainable Development in the curricula in accordance with the document "Guidelines for the Introduction of Sustainability in the Curriculum," approved by the Executive Committee of the CRUE (CRUE refers to the Conference of Presidents of Spanish Universities) Working Group of the Sectorial Commission CRUE-Sustainability. This guide was published in 2005, updated in 2011, and expanded in 2012 [33]. The Sectorial Commission CRUE-Sustainability recommends that four competencies related to sustainability should be included in the curricula of all Spanish university degrees:

- C1: Critical contextualization of knowledge by establishing interrelations with social, economic, environmental, local and/or global problems.
- C2: Sustainable use of resources and prevention of negative impacts on the natural and social environment.
- C3: Participation in community processes that promote sustainability.
- C4: Application of ethical principles related to the values of sustainability in personal and professional behavior.

The competencies defined by the Sectorial Commission CRUE-Sustainability were defined after an exhaustive review of the literature. The objective of this review was to achieve a reduced and complete set of competencies that integrated most of the sustainability competencies identified in different international studies. As can be seen, the four competencies defined by the Sectorial Commission CRUE-Sustainability have a more generic character than the five competencies defined by the Tuning project. Thus, the CRUE C4 competency contains the Tuning C6, C17, C23, and C25 competencies, while the CRUE C2 competency contains the C28 Tuning competency.

The objective of this work is to present a methodology for measuring the level of sustainability presence in an engineering curriculum. Previous studies have presented proposals with similar objectives, but they consisted of proposals that are less generalist. For example, in [13], a methodology was proposed to calculate the relevance ratio index of a curriculum, defined as the relative weight of renewable energy and sustainability topics for energy studies. This methodology was focused on its use in energy degree programs, and some changes are required for its application to other different curricula. The methodology proposed in this work, however, is general and can be used in any engineering degree. In addition, the proposed methodology can be easily applied to other non-engineering degrees simply by changing the sustainability map, as shown in [34].

The work presented in this paper uses a sustainability map as an analytical tool. The validity and reliability of the sustainability map was analyzed in [25]. The map has been designed through an inductive–deductive process based on the four competencies defined by the Sectorial Commission CRUE-Sustainability.

### 1.2. Objectives and Research Questions

The research objectives of this work are the following:

1. To design a methodology and tools to analyze objectively to what extent the competencies related to sustainability are developed in an engineering degree.
2. To apply the methodology to a case study: Ten engineering degrees from three universities.

   a. To analyze the presence of the competencies related to sustainability in the case study using the tools designed in Objective 1.
   b. To compare the presence of competencies related to sustainability between degrees and universities in the case study.

To achieve the previously defined objectives, the aim of this work is to answer the following research questions:

Q1: To what extent is sustainability present in the engineering degrees of the Spanish university system?
Q2: Do Spanish universities have a defined strategy to develop sustainability in their curricula?
Q3: What competencies related to sustainability are present to a greater or lesser extent in the engineering degrees studied?
Q4: Are the competencies related to sustainability present in the different engineering degrees in a uniform way, or are there differences between the degrees?

The results indicate that the sustainability competency least present in all the degrees is C3 (participation in community processes that promotes sustainability), with an average presence of 23.3%, while the most present is C4 (application of ethical principles related to the values of sustainability in personal and professional behavior), with an average presence of 76.6%. The other two competencies, C1 (critical contextualization of knowledge by establishing interrelations with social, economic, environmental, local and/or global problems), and C2 (sustainable use of resources and prevention of negative impacts on the natural and social environment), have an average presence of approximately 55%. As can be seen from these results, the four sustainability competencies are developed very differently in engineering degrees.

In general, learning outcomes related to sustainability have an average presence of 52.1%. Therefore, it can be concluded that practically half of the cells in the ten engineering sustainability maps are not developed in the engineering degrees analyzed.

## 2. Materials and Methods

### 2.1. Methodology

The proposed methodology used two tools:

- The engineering sustainability map
- The sustainability presence map

The engineering sustainability map has been developed within the framework of the EDINSOST (EDINSOST is a project funded by the Ministry of Economy and Competitiveness of Spain, aimed at training professionals in Spanish universities as agents of change to meet the challenges of society.) project [25,35]. The sustainability map is a competency map designed on the basis of the four

competencies related to sustainability defined by the Sectorial Commission CRUE-Sustainability cited in Section 1.

A competency map is a matrix of learning outcomes organized according to competency units (rows) and domain levels (columns) [26]. For each of the four competencies related to sustainability (C1–C4), the rows of the matrix define a set of competency units (different aspects to be dealt with by each competency). Each competency unit is defined in the cells of the matrix through a set of learning outcomes. Learning outcomes are classified in domain levels using a learning taxonomy. The definition of a taxonomy allows the learning process to be sequenced. In the case of the EDINSOST project, a simplified version of Miller's pyramid [36] was used as a taxonomy. This taxonomy has only three levels: L1: Know; L2: Know-how; and L3: Demonstrate and do (the two upper levels of the Miller's pyramid, demonstrate and do, have been integrated into a single level).

Given that sustainability is traditionally studied from the point of view of its three dimensions (economic, social, and environmental), the EDINSOST project follows this approach for analyzing each competency. The holistic dimension is also incorporated when the three dimensions are approached together. The competency units defined for each of the four competencies and the dimension of sustainability in which they are framed are presented in Table 1. With the aim of achieving a reduced (and manageable) number of learning outcomes within the engineering sustainability map, the four competencies were analyzed from the holistic point of view whenever possible.

**Table 1.** Competency units and sustainability dimension related to each competency.

| Competency | Dimension | Competency Units |
|---|---|---|
| C1: Critical contextualization of knowledge | Holistic | C1.H.1. Has a historical perspective (state of the art) and understands social, economic and environmental problems, both locally and globally. |
| | | C1.H.2. Is creative and innovative. Is able to see the opportunities offered by engineering to contribute to the development of more sustainable products and processes. |
| C2: Sustainable use of resources | Holistic | C2.H.1. Takes into account sustainability in his/her work as an engineer. |
| | Environmental | C2.EV.1. Takes into account the environmental impact of his/her work as an engineer. |
| | Social | C2.S.1. Takes into account the social impact of his/her work as an engineer. |
| | Economical | C2.EC.1. Is capable of successfully carrying out the economic management of an engineering project. |
| C3: Participation in community processes | Holistic | C3.H.1. Identifies when the sustainability of a project can be improved if it is done through community collaborative work. Responsibly performs collaborative work related to sustainability. |
| C4: Application of ethical principles | Holistic | C4.H.1. Behaves according to the deontological principles related to sustainability. |

As can be seen in Table 1, all competency units are defined for the holistic dimension, with the exception of the competency C2 (sustainable use of resources), which defines competency units for the four sustainability dimensions. The complete engineering sustainability map [25] can be found in Appendix A.

The sustainability presence map of an engineering degree was drawn up from a quantitative analysis of how C1–C4 sustainability competencies are developed in an engineering degree. The fundamental indicator of this analysis is the number of subjects developing each competency in the degree. The objective of the methodology was not to analyze how the subjects develop the competencies, nor how many hours each subject devotes to each competency, since this implies

personally interviewing all the teachers involved, which is not always possible, especially when the intention is to conduct a broad-spectrum study. For this reason, when building the sustainability presence map of the degree, the methodology consisted in reviewing in depth the learning guides of all the degree subjects. The assigned competencies, objectives, and contents, as well as the activities carried out by the students, were reviewed for each subject. All these aspects have been compared with the learning outcomes of the engineering sustainability map in order to detect any coincidences. It was not enough to search for keywords within the learning guides, since in many cases the teachers who write the guides are not experts in sustainability and do not use a precise technical vocabulary. Learning guides should therefore be analyzed by sustainability experts who know the engineering sustainability map well. In this work, the review of the learning guides was carried out by members of the EDINSOST project. When the subject analysis was carried out on a degree close to the researchers, this information could be completed with data provided by the professors who teach these subjects. The objective of the curriculum analysis was to compare, for each of the sustainability competencies C1–C4, whether the competency was present or not in the objectives, contents and activities of each subject, regardless of their presence level.

The sustainability presence map of an engineering degree is an engineering sustainability map in which the cells corresponding to the learning outcomes contained a number greater than or equal to zero. A 0 indicated that none of the learning outcomes of the cell are developed in the degree. A number greater than zero indicates the number of subjects that develop any of the learning outcomes of the cell. If a cell of the sustainability presence map contained a number greater than zero, it was assumed that both the competency unit and the competency related to the cell were developed in the degree at the domain level in which the cell was located (regardless of the number of subjects or hours dedicated to this development).

To build the sustainability presence map of an engineering degree, the number of degree subjects developing each cell had to be analyzed. Since each cell may have contained several learning outcomes, the number of subjects developing a cell of the map was the sum of the number of different subjects that developed some of the learning outcomes of the cell.

Finally, to achieve Objectives 2.1 and 2.2, we applied the methodology to ten engineering degrees of the Spanish university system, and we studied what competencies and competency units were both more and less present in each degree, as well as the total number of subjects involved.

### 2.2. Case of Study

#### 2.2.1. Sample

In order to carry out a non-exhaustive exploratory study of the engineering degrees of the Spanish university system, a small sample (ten curricula) of three universities was analyzed: The University of Córdoba (UCO), The Universitat Politècnica de Catalunya–BarcelonaTech (UPC) and the Universidad Politécnica de Madrid (UPM). The ten curricula belonged to six different degrees:

- The Bachelor Degree in Electrical Engineering at the UCO and UPC.
- The Bachelor Degree in Informatics Engineering at the UCO, UPC, and UPM.
- The Bachelor Degree in Mechanical Engineering at the UCO and UPC.
- The Bachelor Degree in Design Engineering at UPC.
- The Bachelor Degree in Chemical Engineering at UPM.
- The Bachelor Degree in Industrial-Technologies Engineering at the UPM.

The purpose of this work was not the generalization of the results obtained. Rather, it was to obtain relevant information on how the sustainability competencies are being developed in a set of degrees at this particular time. Though the results are not generalizable, they may indicate how the engineering degrees in the Spanish university system develop these competencies.

### 2.2.2. Research Design

The design used to carry out the present study was quantitative and correlational; it was quantitative insofar as the engineering sustainability map was used to analyze the presence of the sustainability competencies in each of the studied degrees, and it was correlational because the results obtained were compared for the different degrees using descriptive statistics.

With regard to the temporal dimension, it was a transversal design in which the different variables were studied in a single time period: The 2018–2019 course.

## 3. Results and Discussion

### 3.1. Q1: To What Extent Is Sustainability Present in the Engineering Degrees of the Spanish University System?

From the sustainability presence maps of all the analyzed degrees, a single table was constructed to perform a joint analysis of the data. The analysis of these data allowed us to answer the first research question: "To what extent is sustainability present in the engineering degrees of the Spanish university system?"

Table 2 shows the emerging mapping of the learning guide analysis of the ten engineering degrees studied. The table shows, for each degree of each university, the number of subjects developing each cell of the sustainability presence map. The information is classified from the competency units identified in Table 1 and their domain levels (L1 = Know, L2 = Know-how, L3 = Demonstrate and do).

**Table 2.** Number of subjects developing each domain level of each competency unit, classified by university and degree.

| Degree | BDEE | | BDIE | | | BDME | | BDDE | BDCHE | BDITE |
|---|---|---|---|---|---|---|---|---|---|---|
| University | UCO | UPC | UCO | UPC | UPM | UCO | UPC | UPC | UPM | UPM |
| **C1:** Critical contextualization of knowledge by establishing interrelations with social, economic, environmental, local and/or global problems. | | | | | | | | | | |
| C1.H.1　L1 | 0 | 6 | 0 | 0 | 1 | 0 | 3 | 3 | 0 | 0 |
| C1.H.1　L2 | 0 | 3 | 0 | 5 | 1 | 0 | 2 | 3 | 2 | 2 |
| C1.H.1　L3 | 0 | 2 | 6 | 4 | 1 | 0 | 1 | 2 | 2 | 2 |
| C1.H.2　L1 | 0 | 1 | 0 | 0 | 2 | 0 | 0 | 2 | 0 | 0 |
| C1.H.2　L2 | 0 | 0 | 19 | 3 | 1 | 0 | 0 | 2 | 2 | 2 |
| C1.H.2　L3 | 0 | 0 | 0 | 3 | 1 | 0 | 0 | 1 | 3 | 3 |
| **C2:** Sustainable use of resources and prevention of negative impacts on the natural and social environment. | | | | | | | | | | |
| C2.EV.1　L1 | 0 | 2 | 0 | 3 | 0 | 0 | 2 | 2 | 1 | 1 |
| C2.EV.1　L2 | 0 | 1 | 0 | 5 | 2 | 0 | 2 | 4 | 2 | 2 |
| C2.EV.1　L3 | 2 | 1 | 0 | 4 | 2 | 2 | 2 | 4 | 2 | 2 |
| C2.S.1　L1 | 0 | 1 | 0 | 3 | 8 | 0 | 1 | 5 | 0 | 0 |
| C2.S.1　L2 | 0 | 0 | 0 | 4 | 5 | 0 | 0 | 5 | 3 | 3 |
| C2.S.1　L3 | 0 | 0 | 0 | 5 | 5 | 0 | 0 | 5 | 2 | 2 |
| C2.EC.1　L1 | 0 | 0 | 0 | 1 | 1 | 0 | 0 | 0 | 2 | 2 |
| C2.EC.1　L2 | 0 | 0 | 0 | 0 | 1 | 0 | 0 | 0 | 1 | 1 |
| C2.EC.1　L3 | 0 | 0 | 0 | 1 | 0 | 0 | 0 | 0 | 2 | 2 |
| C2.H.1　L1 | 0 | 2 | 0 | 3 | 0 | 0 | 0 | 2 | 1 | 1 |
| C2.H.1　L2 | 0 | 2 | 0 | 1 | 0 | 2 | 0 | 2 | 5 | 5 |
| C2.H.1　L3 | 0 | 1 | 0 | 1 | 0 | 1 | 0 | 1 | 3 | 3 |
| **C3:** Participation in community processes that promote sustainability. | | | | | | | | | | |
| C3.H.1　L1 | 0 | 0 | 0 | 1 | 1 | 0 | 0 | 1 | 0 | 0 |
| C3.H.1　L2 | 0 | 0 | 0 | 1 | 0 | 0 | 0 | 1 | 0 | 0 |
| C3.H.1　L3 | 0 | 0 | 0 | 1 | 0 | 0 | 0 | 1 | 0 | 0 |

**Table 2.** *Cont.*

| Degree | BDEE | | BDIE | | | BDME | | BDDE | BDCHE | BDITE |
|---|---|---|---|---|---|---|---|---|---|---|
| University | UCO | UPC | UCO | UPC | UPM | UCO | UPC | UPC | UPM | UPM |
| **C4:** Application of ethical principles related to the values of sustainability in personal and professional behavior. | | | | | | | | | | |
| C4.H.1    L1 | 0 | 4 | 0 | 2 | 3 | 0 | 2 | 2 | 3 | 3 |
|    L2 | 0 | 3 | 18 | 2 | 3 | 2 | 1 | 2 | 1 | 1 |
|    L3 | 0 | 2 | 16 | 1 | 3 | 0 | 0 | 2 | 1 | 1 |
| Different subjects | 3 | 10 | 36 | 13 | 11 | 4 | 3 | 9 | 7 | 7 |

The columns in Table 2 show the analyzed degrees organized by university:

- Universities: University of Córdoba (UCO), Universitat Politècnica de Catalunya-BarcelonaTech (UPC), and Universidad Politécnica de Madrid (UPM).
- Degrees: Bachelor Degree in Electrical Engineering (BDEE), Bachelor Degree in Informatics Engineering (BDIE), Bachelor Degree in Mechanical Engineering (BDME), Bachelor Degree in Design Engineering (BDDE), Bachelor Degree in Chemical Engineering (BDCHE), and Bachelor Degree in Industrial-Technologies Engineering (BDITE).

The rows in Table 2 contain the number of different subjects developing each competency unit in each of the three domain levels considered.

For example, the cell corresponding to the BDEE of the UCO and row L1 of the competency unit C1.H.1. indicate the domain level L1 of the competency unit "Has a historical perspective (state of the art) and understands social, economic and environmental problems, both locally and globally." This cell contains the value 0, which means that no subject of the BDEE of the UCO develops this competency unit in the domain level L1. On the other hand, the BDEE of the UPC presents a 6 in this same row, indicating that six subjects develop the domain level L1 of the competency unit C1.H.1.

The engineering sustainability map of an engineering degree is composed of 24 cells, which in Table 2 are presented in vertical format under the acronym of each degree. Since 47.1% of the cells in Table 2 (113/240) contain the value 0, it can be concluded that at least 47.1% of the learning outcomes expected in sustainability are not developed in the engineering degrees analyzed. This number is likely to be even higher, since each cell of the engineering sustainability map may contain more than one learning outcome, and if any of them are developed in any subject, the value of the cell is different from 0, although some learning outcomes of the cell are not developed in any subject of the degree.

Table 2 also shows, in the last row, the total number of different subjects that develop sustainability in each degree of each university. This number is not the sum of the values of each column, since some subjects develop several domain levels of different competency units; therefore, the subjects can be repeated in different cells. For example, a subject X of the UPC BDEE could develop domain levels L1, L2 and L3 of the competency unit C1.H.1. This subject should be considered only once for counting the number of subjects that develop sustainability in the degree, but if the row "different subjects" was the result of the sum of the corresponding columns, the subject X would be counted three times in this case.

As can be seen, the number of different subjects varies greatly, even for the same degree taught in different universities. For example, only three subjects of the BDEE of the UCO develop sustainability, while 10 subjects develop it in the BDEE of the UPC.

Figure 1 shows these data graphically. For the degrees that are taught in more than one university, both the average number of subjects that develop sustainability and the standard deviation are shown. In the case of the UCO, data are shown with and without the BDIE, since this degree can be considered as an outlier when comparing the number of subjects that develop sustainability in this degree with that of the other degrees. For the degrees that are taught in a single university, the number of subjects is indicated directly, with a deviation type 0. In the four bars on the right of the figure, the average number of subjects of the ten degrees and their standard deviation are shown with and without the

BDIE of the UCO. For each degree, the bar on the left indicates the number of subjects (or its average), and that on the right shows the degree of dispersion, with respect to the average, that occurs in the data. As may be seen in figure, the BDIE is the degree with the greatest dispersion of data (36 subjects in the UCO vs 11 subjects in the UPM). This dispersion is drastically reduced when the BDIE of the UCO is not considered.

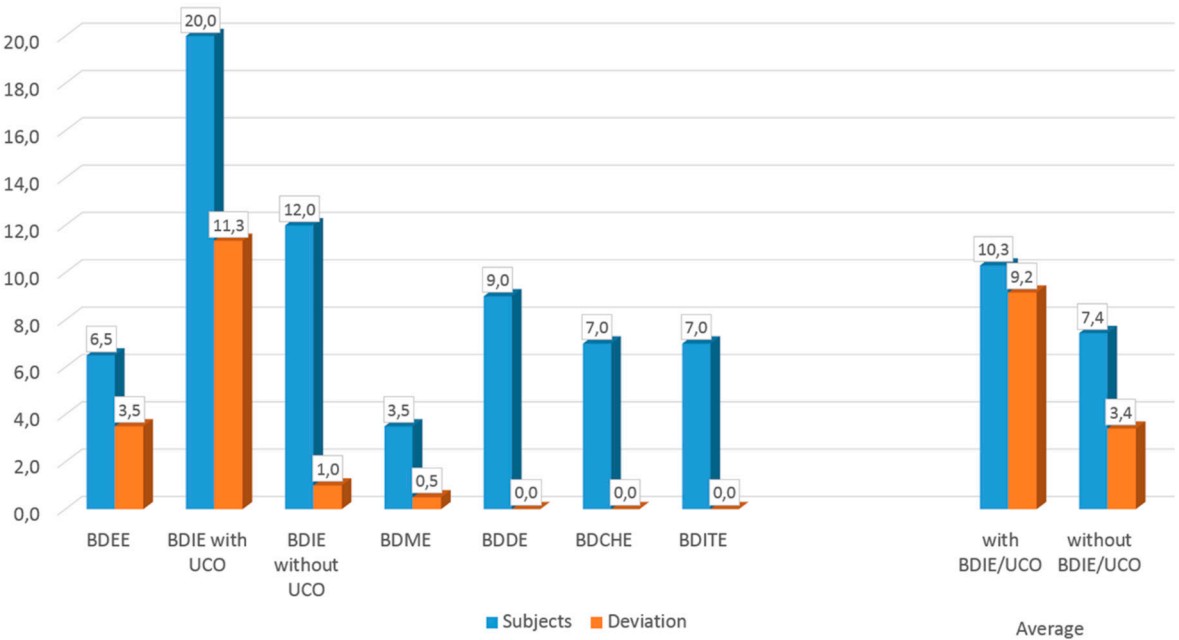

**Figure 1.** Average of subjects that develop sustainability in each degree, and standard deviation.

*3.2. Q2: Do Spanish Universities Have a Defined Strategy to Develop Sustainability in Their Curricula?*

Figure 2 shows the same information as Figure 1, but here it is organized by universities.

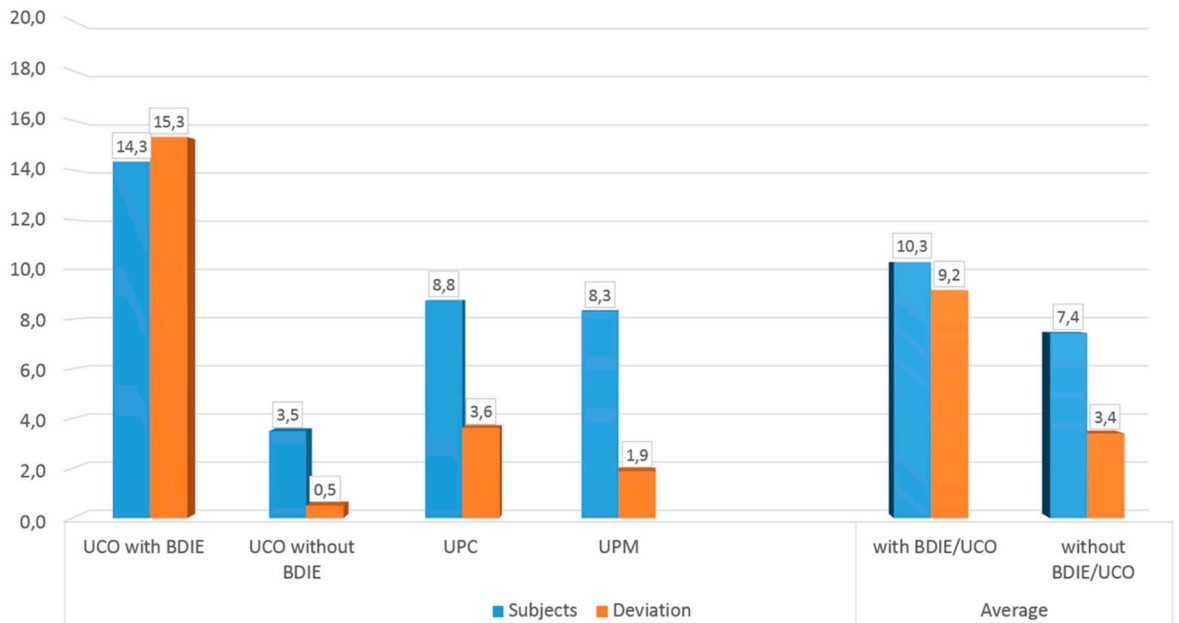

**Figure 2.** Average of subjects that develop sustainability in each university, and standard deviation.

In the case of the UCO, the data are presented with and without the BDIE. On the right, the mean and standard deviation of the degrees are presented again, with and without the BDIE of UCO. The figure shows that the average number of subjects, when the BDIE of the UCO is not considered, is 7.4, with a standard deviation of 3.4.

Not all universities have the same policy about including sustainability in their curricula, and several obstacles and limitations hinder the introduction of sustainability in the curricula of their degrees. Some limitations are internal, due to the culture and structure of each university, while others are imposed by external factors, such as the lack of social pressure and/or—unlike what happens in other countries of its environs—the absence in Spain of a clear state or autonomous community strategy to support the application and development of sustainability policies in higher education [37]. Unfortunately, for this reason, the level of commitment that a university acquires with sustainability in Spain depends on initiatives that, in many cases, are thanks to professors or workers at the university (but not by the institution itself). These initiatives are usually endowed with few human and economic resources. Figures 1 and 2 clearly show this situation. The number of subjects that each degree/university devotes to developing sustainability is highly variable. For example, the data in Table 2 reveals that the same degree may show a significant difference in the number of subjects that develop sustainability, depending on the university. For instance, the BDEE is developed in three subjects in the UCO and in 10 in the UPC, while the BDIE is developed in 36 subjects in the UCO, 13 in the UPC, and 11 in the UPM. This difference results in a high standard deviation when the mean is calculated.

On the other hand, in general, it seems that no defined university strategy exists regarding the number of subjects that should develop sustainability in their curricula, as shown in Table 2 and Figure 2. The UCO is a clear example: Sustainability is developed in three subjects in the BDEE (the lowest number of subjects of the 10 studied degrees, together with the BDME of the UPC), while it is developed in the BDIE in 36 subjects (the highest number of subjects studied). It appears that no strategy exists on the part of the UCO regarding the number of subjects that must develop sustainability in each degree. The number of subjects dedicated to developing sustainability at the UPC ranges from three to 13, so there seems to be no defined strategy here either. At the UPM, however, the two degrees from the industrial school develop sustainability in seven subjects, and the computer school develops sustainability in 11 subjects. This is compatible with the existence of a certain school strategy but not with a university strategy. In all cases, we assume that the number of subjects that develop sustainability in each university in any engineering degree should be similar, since all engineering degrees have a common sustainability map. Two different universities may have different strategies and may devote a different number of subjects, but within each university, this number would have a certain homogeneity if the university had a strategy on sustainability.

*3.3. Q3: What Competencies Related to Sustainability Are More and Less Present in the Engineering Degrees Studied?*

Table 3 synthetically summarizes the information presented in Table 2. This table allows us to answer the third research question: "What competencies related to sustainability are more and less present in the engineering degrees studied?"

Table 3 shows the following data for each domain level of each competency unit:

- The number (N) of different engineering degrees in which the competency is developed at that domain level. As can be seen, N ≠ 0 in all cases, since none of the rows in Table 2 contain only zeros. From the data in Table 3, it appears that at least two degrees of those studied (note that the minimum value of N is N = 2) are developing some learning result of any domain level for any competency.

- The percentage of engineering degrees that do not develop any learning outcome of that level (% not LO). For example, the domain level L1 of the competency unit C1.H.2 (is creative and innovative) is not present in 70% of the degrees, since only three of the ten degrees studied (N = 3) develop learning outcomes in this domain level (the BDEE of the UPC, the BDIE of the UPM, and the BDDE of the UPC, as shown in Table 2).
- The percentage of map cells, corresponding to a competency unit, that are not developed by any engineering degree (% not CU, percentage of cells containing zeros in each competency unit in Table 2).
- The percentage of map cells, corresponding to a competency, that are not developed by any engineering degree (% not C, percentage of cells containing zeros in each competency in Table 2).

**Table 3.** Analysis of the non-presence of competencies in sustainability in the engineering degrees.

|  | Level | N | % not LO | % not CU | % not C |
|---|---|---|---|---|---|
| **C1:** Critical contextualization of knowledge by establishing interrelations with social, economic, environmental, local and/or global problems. | | | | | |
| C1.H.1 | L1 | 4 | 60 | | |
|  | L2 | 7 | 30 | 36.67 | |
|  | L3 | 8 | 20 | | 45 |
| C1.H.2 | L1 | 3 | 70 | | |
|  | L2 | 6 | 40 | 53.33 | |
|  | L3 | 5 | 50 | | |
| **C2:** Sustainable use of resources and prevention of negative impacts on the natural and social environment. | | | | | |
| C2.EV.1 | L1 | 6 | 40 | | |
|  | L2 | 7 | 30 | 26.67 | |
|  | L3 | 9 | 10 | | |
| C2.S.1 | L1 | 5 | 50 | | |
|  | L2 | 5 | 50 | 50 | |
|  | L3 | 5 | 50 | | 46.67 |
| C2.EC.1 | L1 | 4 | 60 | | |
|  | L2 | 3 | 70 | 66.67 | |
|  | L3 | 3 | 70 | | |
| C2.H.1 | L1 | 5 | 50 | | |
|  | L2 | 6 | 40 | 43.33 | |
|  | L3 | 6 | 40 | | |
| **C3:** Participation in community processes that promote sustainability. | | | | | |
| C3.H.1 | L1 | 3 | 70 | | |
|  | L2 | 2 | 80 | 76.67 | 76.67 |
|  | L3 | 2 | 80 | | |
| **C4:** Application of ethical principles related to the values of sustainability in personal and professional behavior. | | | | | |
| C4.H.1 | L1 | 7 | 30 | | |
|  | L2 | 9 | 10 | 23.33 | 23.33 |
|  | L3 | 7 | 30 | | |

For example, the competency C3 (participation in community processes), with a single competency unit (C3.H.1), is the competency with the least presence in the engineering degrees studied (76.7% of non-presence). This competency is developed only in three universities in the domain level L1 and in two universities in the domain levels L2 and L3 (only seven of the 30 cells of the competency contain numbers greater than zero, see Table 2). In contrast, competency C4.H.1 (behaves according to the deontological principles) is the competency with the highest presence, with a degree of non-presence of 23.3%.

The data presented in Table 3 show that the competency most present in the engineering degrees is the C4 (application of ethical principles) with only 23.3% of non-presence, while the least present is C3 (participation in community processes) with 76.7% of non-presence. Competencies C1 (critical contextualization of knowledge) and C2 (sustainable use of resources) are not present by 45% and 46.7%, respectively. These competencies therefore have an average presence in the studied degrees of approximately 55% and 53%. Assuming that the weight of the four competencies on the sustainability learning is the same, we can conclude that, on average, the presence of sustainability in the curricula is 52.1% (arithmetic mean of the presence of the four competencies).

It is encouraging to observe how the orientation towards transversal training and towards a civic ethic, both in personal and deontological behavior, which has a long tradition in higher education degrees, is also present in the analyzed engineering degrees, as may be deduced from the high presence of the competency C4 (application of ethical principles) in the engineering sustainability map. This fact is especially valuable because many engineering degrees devote much more time to developing economic and environmental aspects than social and ethical aspects.

It is probable that the influence of the creation of the European Higher Education Area, where "the university marks new spaces to carry out proposals of ethical learning, connected with new ways of professional and scientific training" [38], is beginning to be noticed in engineering degrees. However, the report prepared for the GUNI (Global University Network for Innovation) [39], in which more than 500 experts from around the world participated, warns that "teaching programs usually contain a hidden agenda of practices based on unsustainable actions that do not favor reflection on the ethical values of human actions in the middle." The integration of sustainability in engineering classrooms will only be possible by transforming the methodologies and evaluation systems of the teaching–learning process. On the other hand, the teacher must discuss the limitations and ethical implications that may arise from the application of certain technological advances with students.

It is worrisome to note the low presence of the competency C3 (participation in community processes) in engineering degrees. Some studies indicate that universities are perceived as disconnected from society and without a direct involvement in its real problems [39]. The studies have also shown that, although universities are active in terms of participation in sustainability issues, the actions carried out are mainly unidirectional from the university to the university community [40]. "Progress towards sustainable universities is only possible if the strategies undertaken are accompanied by measures to involve the different university bodies as well as external agents" [40] in a true participation. This participation should enable future engineers to understand that the sustainability of a technological project will provide better responses to society if the project is carried out in a collaborative, community and reciprocal framework.

Higher education needs its own transformation to be transformative itself and move towards an education for sustainability. The existence of a strategy of change in the entire university institution is necessary, both at the academic level (teaching and research) and at an extra-academic level, in connection with the community which the university is required provide [41,42].

Competencies C1 (critical contextualization of knowledge), and C2 (sustainable use of resources) are clearly those most closely related to the technical aspects of engineering. However, almost half of the cells in the engineering sustainability map are not developed in any subject in the ten degrees analyzed. Moreover, when the competency units related to these competencies are analyzed, the competency unit C1.H.2. (is creative and innovative) is not present in more than half of the degrees analyzed (53.3%). If universities do not train creative and innovative engineers, the move towards a more sustainable world will be much more difficult. Furthermore, if creativity and innovation are not fostered in the university, where will they develop? Creativity, innovation, and the critical contextualization of knowledge are elements that enable the achievement of the common objectives required to face current challenges and undertake complex mental tasks that go beyond the basic reproduction of accumulated knowledge [43].

In a broad sense, engineering consists of the set of scientific and technological knowledge necessary for innovation, invention, development and improvement of techniques and tools to meet the needs and to solve the problems faced by society and companies. Promoting the prospective orientation of alternative scenarios is therefore fundamental in engineering degrees. The prospective orientation of alternative scenarios favors critical thinking and decision making.

It is also worrying to note that the competency unit C2.S.1. (takes into account the social impact of his/her work as an engineer) is not present in 50% of the analyzed degrees. The social aspects are clearly beyond the scope of many of the engineering degrees. However, engineers should be tasked with designing devices that help to improve living conditions in society.

Finally, it is rather surprising that the competency unit C2.EC.1. (is capable of successfully carrying out the economic management of an engineering project) is not present in 66.7% of the analyzed degrees. Engineers trained at the university will work on projects throughout most of their professional life and thus should be capable of managing them—but how will they manage projects if they are not trained to carry them out economically? Clearly, the university is currently delegating this aspect of training to the companies that hire the graduates.

The results presented in this paper are compatible with those found in the study of the Working Group on the Evaluation of University Policies on Sustainability within the framework of the CRUE [44]. This study indicates that Spanish universities show greater progress in actions concerning environmental awareness than in related activities, both through the development of social responsibility programs as well as the evaluation of the economic and environmental impact generated by university activities.

### 3.4. Q4: Are the Competencies Related to Sustainability Present in the Different Engineering Degrees in a Uniform Way, Or Are there Differences between the Degrees?

This work assumes that a competency related to sustainability is present in a given degree when any of the learning outcomes of the competency is developed in any degree subject.

For each degree of each university, a fractional value between 0 and 1 was assigned to each of the four C1–C4 competencies. This value, which indicates the percentage of presence of the competency in the degree, was calculated from the data shown in Table 2, depending on the number of domain levels that are developed in all competency units. For example, "2/3" is the value assigned to C4 (application of ethical principles) in the BDIE of the UCO because two of the three domain levels are developed: The L2—know-how (18 subjects) and L3—demonstrate and do (16 subjects) levels are developed, but the L1—know level (0 subjects) is not developed. For C1 (contextualization of knowledge) and C2 (sustainable use of resources), which have more than one competency unit, all levels of all competency units must be considered. For example, "1/12" is the value assigned to C2 (sustainable use of resources) in the BDEE of the UCO because only the L3 level of the competency unit C2.EV.1 (environmental impact of the work as an engineer) is developed in this degree (one cell of twelve).

These data are shown below in Table 4. The shaded and highlighted cells correspond to the examples described in the previous paragraph. The rows indicate the different degrees analyzed, while the columns show the number of different subjects that develop sustainability in each degree (N), the university (U), and the percentages of presence of the four C1–C4 competencies.

Table 4 enables us to answer the fourth research question: "Are the competencies related to sustainability present in the different engineering degrees in a uniform way, or are there differences between the degrees?"

Figure 3 graphically presents the data in Table 4. The figure shows a comparison of the percentage of average presence of the four sustainability competencies in the six degrees analyzed. The abscissa axis identifies the four competencies C1–C4. For each competency, several bars are displayed (one bar per degree). When a degree is taught in more than one university, the figure presents the average of the data from the universities in which it is taught (identified as "average" in Table 2).

**Table 4.** Percentage of presence of sustainability competencies in the engineering degrees analyzed (range 0–1).

| Degree | N | U | C1 | C2 | C3 | C4 |
|---|---|---|---|---|---|---|
| | 3 | UCO | 0 | 1/12 | 0 | 0 |
| BDEE | 10 | UPC | 2/3 | 7/12 | 0 | 1 |
| | | Average | 0.3333 | 0.3333 | 0 | 0.5 |
| | 36 | UCO | 1/3 | 0 | 0 | **2/3** |
| BDIE | 13 | UPC | 2/3 | 11/12 | 1 | 1 |
| | 11 | UPM | 1 | 7/12 | 1/3 | 1 |
| | | Average | 0.6667 | 0.5 | 0.4444 | 0.8889 |
| | 4 | UCO | 0 | 1/4 | 0 | 1/3 |
| BDME | 3 | UPC | 1/2 | 1/3 | 0 | 2/3 |
| | | Average | 0.25 | 0.2917 | 0 | 0.5 |
| BDDE | 9 | UPC | 1 | 3/4 | 1 | 1 |
| | | Average | 1 | 0.75 | 1 | 1 |
| BDCHE | 7 | UPM | 2/3 | 11/12 | 0 | 1 |
| | | Average | 0.6667 | 0.9167 | 0 | 1 |
| BDITE | 7 | UPM | 2/3 | 11/12 | 0 | 1 |
| | | Average | 0.6667 | 0.9167 | 0 | 1 |

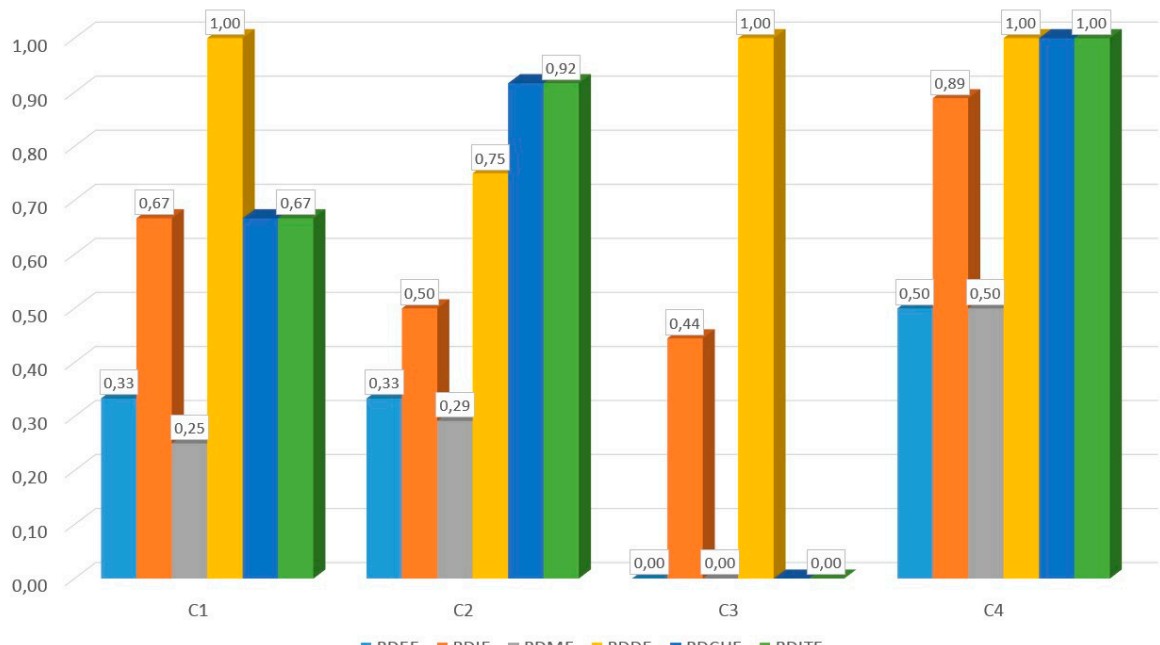

**Figure 3.** Percentage of the average presence of the four sustainability competencies in the six engineering degrees analyzed.

Figure 3 shows that no pattern exists for the competencies; that is, no competency has the same level of presence for all degrees. However, C3 (participation in community processes) is not present in four of the six degrees analyzed. On the other hand, C4 (application of ethical principles) has a 100% presence in three degrees, and C2 (sustainable use of resources) has a 92% presence in two degrees.

Figure 4 presents the same data shown in Figure 3, but they are organized by degree/university rather than by competency. The figure provides a graphic illustration of the degrees that have more sustainability presence.

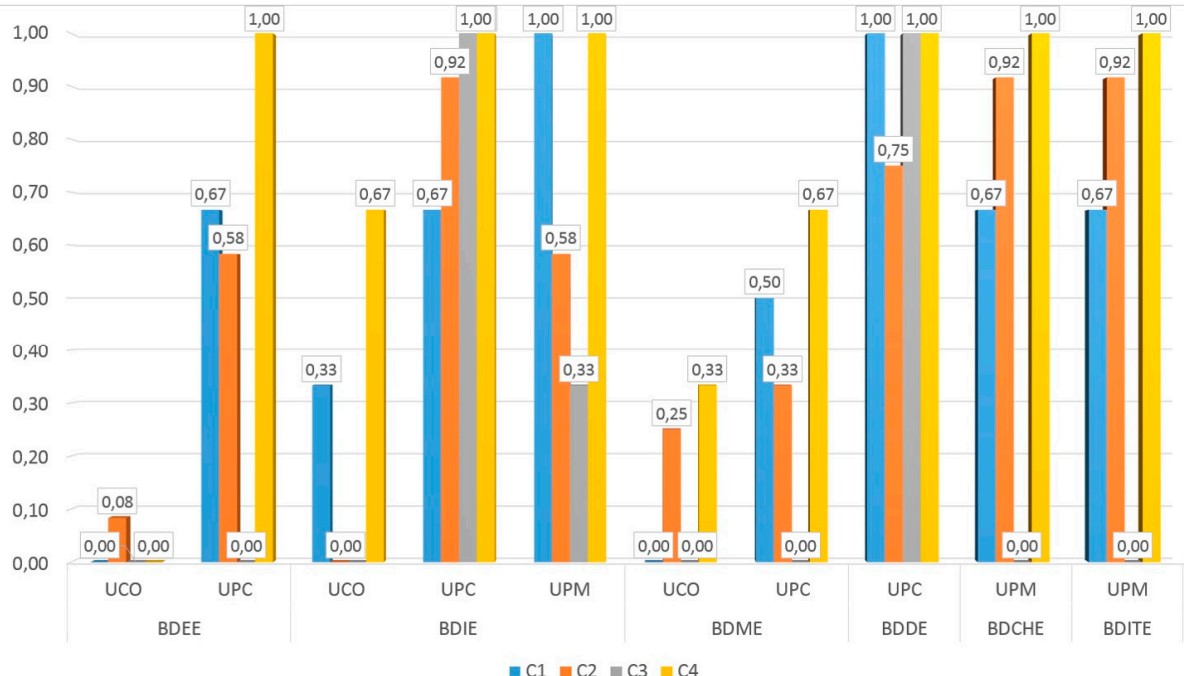

**Figure 4.** Percentage of the presence of sustainability competencies in the engineering degrees studied.

Figure 4 shows that some degrees have several very high bars (high presence of some sustainability competencies), while other degrees have few bars or bars that are very low (little presence of sustainability competencies). UCO's BDEE is the degree with the least presence of sustainability, followed by the BDME (both in UCO and in UPC). On the other hand, the degree that shows more sustainability presence is the BDDE of UPC, with a 100% presence in competencies C1 (critical contextualization of knowledge), C3 (participation in community processes), and C4 (application of ethical principles), and it has a 75% presence of C2 (sustainable use of resources), followed by BDIE (also from UPC), with a 100% presence in competencies C3 (participation in community processes) and C4 (application of ethical principles), 92% in C2 (sustainable use of resources) and 67% in C1 (critical contextualization). This result is not surprising, given that this degree has been designed according to a strategy for developing sustainability through several curriculum subjects [26] and which uses the engineering sustainability map as a tool. This is not the case with the BDDE, which obtains however better results. With respect to the UPM, two of its degrees (BDCHE and BDITE, on the right in the figure) are taught in the same school, and the results are identical because the school's sustainability strategy is the same for both degrees (unlike the BDIE, which is taught by a different school).

Figure 5 groups the data in Figure 4 to graphically show the level of contribution of each competency to the Education for Sustainability in each degree. Degrees are classified by university on the abscissa axis. The level of presence of sustainability for each degree/university is presented on the ordinate axis. The level of presence, considered as a number between 0 and 4, is the sum of the percentage (between 0 and 1) of presence of each of the four sustainability competencies in each degree/university. Each competency is represented by a different color to clearly identify its contribution to the sustainability presence in each degree.

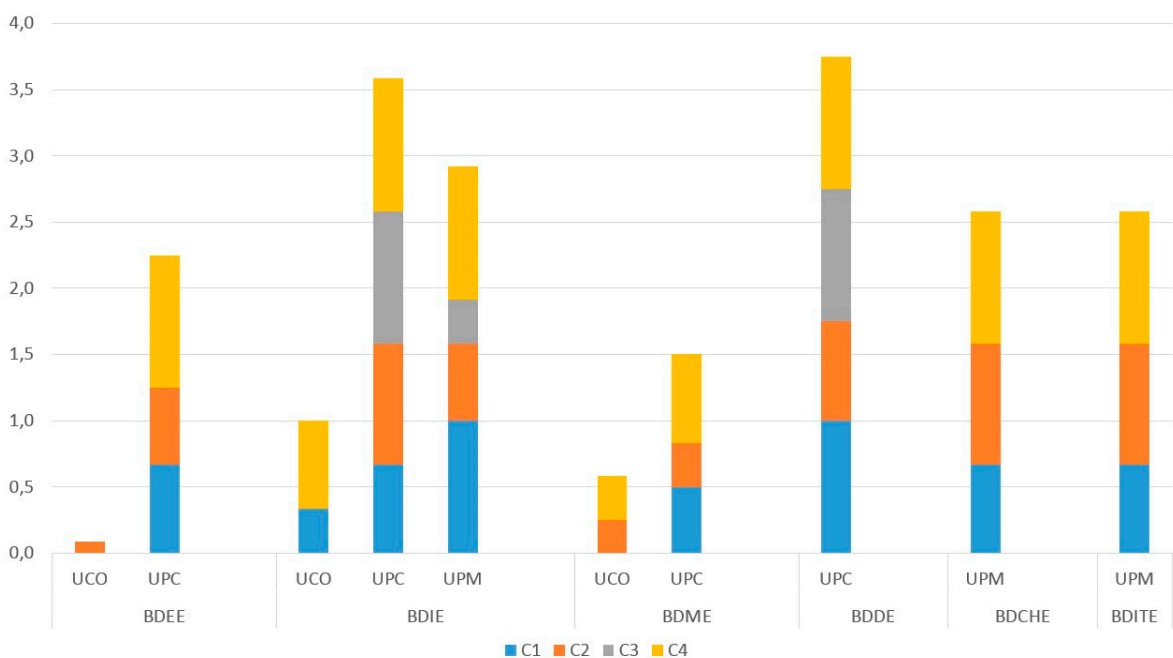

**Figure 5.** Level of sustainability presence for each university/degree, in a range of 0–4.

Figure 5 clearly shows that the degrees with the greatest sustainability presence are the BDDE and the BDIE of UPC, and the one with the least presence is BDEE of UCO, which only develops (and little) the competency C2 (sustainable use of resources). This figure provides valuable information regarding how sustainability is treated in the different universities.

It is as interesting as knowing what competencies are developed in a degree as it is to know in which domain level this development is carried out. Then, an analysis of the presence of the four competencies was made in lines below from the perspective of the three domain levels of the taxonomy. Figure 6 shows, for each of the three domain levels of the four sustainability competencies (know, know how, demonstrate and do), the average presence of each domain level in the ten analyzed curriculums.

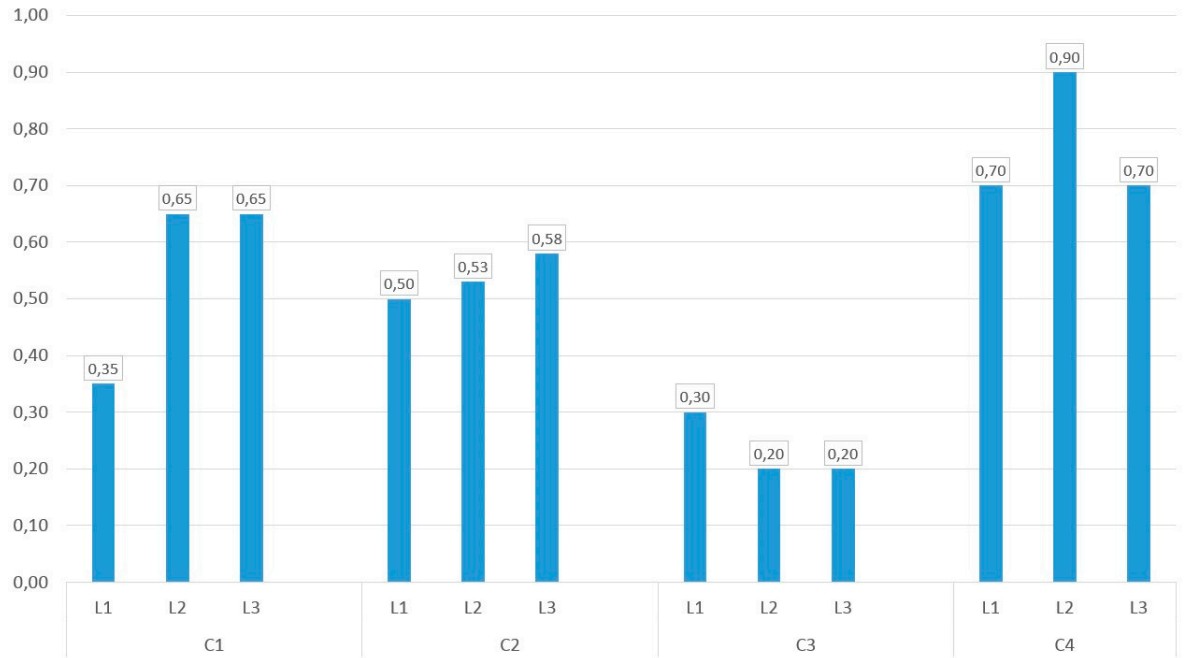

**Figure 6.** Average presence of each domain level in the ten engineering degrees studied.

Figure 6 clearly shows how, with the exception of the competency C3 (participation in community processes), level L1 (the lowest of the taxonomy) is the least present. The conclusion is that the degrees directly develop the highest levels of the taxonomy, devoting little time to the level "know." This is probably because the aim of engineering curricula is often that future engineers should seek solutions to problems based on the application of technological advances without having knowledge of the social, economic and/or environmental impact that may result from their actions at work [12].

In the case of competency C1 (critical contextualization of knowledge), the level "know" has an approximate 35% presence, compared to 65% of the levels "know-how" and "demonstrate and do." This fact is especially striking, considering that the two related competency units are C1.H.1. (historical perspective—state of the art) and C1.H.2. (is creative and innovative). While it could be understood that the competency unit C1.H.2. has a greater presence in the highest levels of the taxonomy, it seems reasonable that C1.H.1. would have a much greater presence at the lowest level. Quite the opposite happens with the competency unit associated with C3 (participation in community processes), C3.H.1. (identify when the sustainability of a project can be improved if it is done through community collaborative work), which have a greater presence from the lowest level of the taxonomy. The fact that this competency has the least presence in engineering degrees, in addition to the fact that the highest levels of the taxonomy are the least developed, suggests that the teaching staff does not possess this competency or does not consider that it should be developed in an engineering degree.

We also considered it interesting to determine how many degrees develop each of the domain levels of the four competencies. Table 5 shows this information, extracted from the data in Table 2. In the table, a color code has been used to clearly identify the domain levels of each competency developed by a greater or a lesser number of degrees. The domain levels developed in four or fewer degrees are shown in red; those developed between five and seven degrees are shown in orange, and those developed in eight or more degrees are shown in green.

**Table 5.** Number of degrees that develop each of the domain levels of each competency.

| | | L1 KNOW | L2 KNOW HOW | L3 DEMONSTRATE and DO |
|---|---|---|---|---|
| **C1:** Critical contextualization of knowledge | C1.H.1. Historical perspective | 4 | 7 | 8 |
| | C1.H.2. Creative and Innovative | 3 | 6 | 5 |
| **C2:** Sustainable use of resources and prevention of negative | C2.EV.1. Environmental impact | 6 | 7 | 9 |
| | C2.S.1. Social impact | 5 | 5 | 5 |
| | C2.EC.1. Economic management of an engineering project | 4 | 3 | 3 |
| | C2.H.1. Consider sustainability in the work as an engineer | 5 | 6 | 6 |
| **C3:** Participation in community | C3.H.1. Sustainability improved by collaborative work | 3 | 2 | 2 |
| **C4:** Application of ethical principles | C4.H.1. Behaves according to deontological principles | 7 | 9 | 7 |

■ 0–4 degrees; ■ 5–7 degrees; ■ 8–10 degrees.

Table 5 shows that only three of the twenty-four cells, 12.5%, are developed by eight or more degrees (green cells). These cells correspond to the L3 level (demonstrate and do) of the competency units C1.H.1. (historical perspective) and C2.EV.1. (environmental impact), and to the L2 level (know how) of the competency unit C4.H.1. (behaves according to deontological principles). It seems reasonable that, as engineers, students should be able to identify the main causes and consequences of a problem related to the sustainability that a product or a service related to engineering can have and are able to relate them to known problems and solutions previously applied ("demonstrate and do" domain level of competency unit C1.H.1.). It is also normal for degrees to consider important "taking into account the environmental effects of the products and services related to engineering"

("demonstrate and do," C2.EV.1.), since a large part of society only identifies sustainability with its environmental dimension. However, it is a pleasant surprise to note that nine of the 10 degrees analyzed are concerned that students are "able to assess the implications of the deontological principles related to sustainability in a project in the field of engineering" ("know-how," C4.H.1.).

With respect to the domain levels developed by fewer degrees, eight cells of the twenty-four (33.3%) are shaded in red. Four of these cells assume, in addition, 50% of the learning outcomes of level L1 (know). Apparently, degrees consider that students should enter the university with a basic training in sustainability and go on to directly develop levels L2 and L3, avoiding the development at the L1 level (note that level L1 is not developed in most of the degrees—not shaded in green—in any of the eight competency units). The other four cells shaded in red belong to the L2 and L3 levels of the competency units C2.EC.1. (economic management of an engineering project) and C3.H.1. (sustainability improved by collaborative work), respectively. While it is understandable that encouraging collaborative work is not one of the current priorities in engineering schools, it is surprising that only approximately 30% of schools develop any of the three domain levels of the competency unit C2.EC.1. (economic management of an engineering project), as already mentioned in previous paragraphs.

Finally, Figure 7 shows the correlation that exists between the variables "number of different subjects that develop sustainability in a degree/university" (axis of abscissas), and "level of presence of sustainability in the degree/university" in the range 0–4 (as presented in Figure 5).

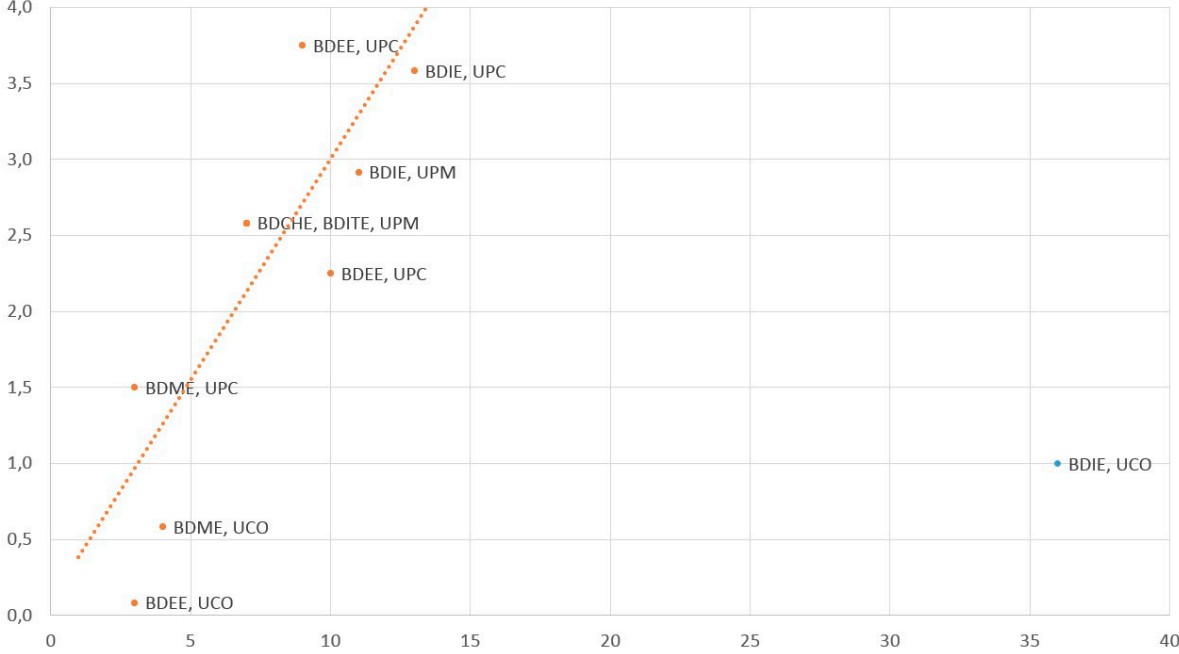

**Figure 7.** Correlation between the variables "number of different subjects that develop sustainability in a degree/university" and "level of presence of sustainability" in the engineering degrees analyzed.

Figure 7 shows the existence of a direct and linear relationship between the two variables studied: Universities with a greater number of subjects developing sustainability are also those that have more presence of the four sustainability competencies. In other words, the academic objectives of the subjects are reasonably distributed among the four sustainability competencies. This may be due to a curriculum design strategy, or it could be due to the different concerns of the professors who teach these subjects. In any case, these data are compatible with the existence of a strategy, on the part of the degrees, to develop sustainability. The point determined by the BDIE of the UCO, to the right of the figure, clearly demonstrates that it is an outlier, as already identified in Section 3.2.

The BDIE of UCO deserves special attention. It is the degree that shows a greater number of subjects developing sustainability (36, according to Table 2), but is one of the degrees that presents a lower sustainability presence. In addition, only two of the four sustainability competencies have some presence in this degree: 33% of presence for C1 (critical contextualization of knowledge) and 67% for C4 (application of ethical principles), according to Figures 4 and 5. Table 2 shows that the degree subjects are concentrated in the competency units C1.H2. and C4.H1. Thus, some specific competencies have been greatly enhanced, although there seems to be no global strategy in the degree to cover the different aspects of sustainability. This example clearly shows the need for a strategy in each degree/university to develop sustainability.

### 3.5. Limitations of This Work and Future Work

This work has some limitations that must be taken into account when considering the results and conclusions presented.

First, the analysis has been conducted on a reduced set of curricula (ten) belonging to only a few universities (three). These curriculums belong to six different degrees. In addition, some of the curricula studied are taught by the same school (two in the case of the UPM). Spain has 87 registered universities and 231 engineering degrees (some of them with very similar names), so the three universities and six degrees analyzed in this work constitute a very small sample. However, the three universities considered belong to the group of universities with a greater number of students and a higher prestige within the Spanish university system. Therefore, although the results presented in this paper cannot be generalized, the authors consider them to be a representative sample of the presence of sustainability in Spanish engineering degrees. The complicity (and willingness) of a large number of teachers from most Spanish universities would be necessary in order to carry out a study with any claim to statistical validity. In addition, these teachers should be trained to analyze the learning guides of the subjects and relate their content to the learning outcomes of the engineering sustainability map. We believe that it is very difficult for these conditions to be met in the short term, so it is important to have studies of this nature which, while they may not ensure statistical validity, would allow us to acquire a vision of the presence of sustainability in some engineering degrees. The objective of the EDINSOST2 project (2019–2022), a continuation of the EDINSOST project (2016–2019), is to improve the sustainability training of teachers of the Spanish university system, thereby enabling them to use the tools designed by EDINSOST in their degrees.

Secondly, the definition of "presence" used in this paper has a limited meaning, since quantifying concepts that have a strong qualitative component is complicated. Presence is treated in this paper as a Boolean variable. For a given cell, it is known whether or not there is a presence but not how much this presence may be. A cell in the engineering sustainability map may contain several learning outcomes. However, it is sufficient for a single learning outcome of the cell to be developed in a single subject of the degree in order to define the whole presence of the cell. In addition, the cell corresponds to a single domain level of a certain competency unit, although the fact that the cell is considered as "present" leads to both the presence of the competency unit and the presence of the related sustainability competency. That is, developing a single learning outcome results in the determination of the presence of a competency and a competency unit in the degree, regardless of the number of subjects and hours that the degree dedicates to developing the learning outcome.

Thirdly, the analysis presented in this paper has been conducted on the basis of the degrees learning guides; only in some very dubious cases have the teachers of the subjects been consulted. Therefore, activities different from those indicated in the learning guides could be undertaken in the subjects.

Despite all the existing limitations, the results presented are compatible with those provided by other studies conducted in Spain [44]. We therefore believe that the data presented in this paper may contribute to a knowledge of the sustainability presence in the Spanish engineering degrees and thus assist in tackling the challenge of integrating sustainability in the degrees in a holistic manner.

This paper has presented a quantitative analysis of the sustainability presence in engineering degrees. The results obtained must be completed by carrying out a qualitative analysis. With this objective, the EDINSOST project has designed a questionnaire for the final year students of an engineering degree. The questionnaire consists of 34 questions directly related to the learning outcomes of the engineering sustainability map [45]. The questions are formulated in the form of statements that students must answer according to a Likert scale of four points [46], according to their level of agreement or disagreement with the statement. The results of the questionnaire will allow knowing, for each engineering degree, the student's perception of the level of acquisition (qualitative information) of each learning outcome, each competency unit, and each competency of the engineering sustainability map.

The questionnaire has already been submitted to and completed by students in all the engineering degrees mentioned in this work, and the data analysis is currently being carried out. The results obtained in each engineering degree will be compared with the sustainability presence map of the degree. This comparison will allow us to know if a correlation exists between the presence of sustainability in an engineering degree and the perception of students' sustainability learning.

The authors consider that it is necessary to advance in this direction by measuring not only the presence of sustainability in the curricula but also the level of graduate training. This information is essential to determine how sustainability is being developed in the different university degrees. From this diagnosis, curriculum designers can identify the weaknesses of current curricula and include learning outcomes that are not developed or not developed enough. Such a diagnosis would also reveal the areas in which the teachers need more training and also enable the design of a training program aimed at covering the needs of each degree. The engineering sustainability map provides the perfect tool for guiding these objectives.

## 4. Conclusions

The general objective of this work was to determine to what extent sustainability is present in the engineering degrees of the Spanish university system. To this end, ten engineering degree learning guides from three Spanish universities were analyzed. It is possible to detect from these learning guides which subjects develop sustainability in each degree.

The instruments used to perform the analysis are the engineering sustainability map and the sustainability presence map. The engineering sustainability map is a matrix drawn up from the four sustainability competencies defined by the CRUE [33]. The cells of the matrix contain the learning outcomes that are related to each of the four sustainability competencies that engineering graduates should have acquired at the end of their studies. These learning outcomes were classified using a learning taxonomy of three domain levels. The sustainability presence map is an engineering sustainability map in which, instead of learning outcomes, the cells contain an integer greater than or equal to zero. A 0 indicates that none of the learning outcomes of the cell are developed in the degree. A number greater than zero states the number of subjects that develop any of the learning outcomes of the cell.

The subjects of the learning guides of the ten engineering degrees studied in this paper have been analyzed to see if they develop any of the learning outcomes of the engineering sustainability map. Based on this information, the sustainability presence map of each engineering degree was designed. From the sustainability presence map, it was possible to calculate the presence of each of the sustainability competencies in the curriculum of each degree.

The findings of this research show that the least present competency in engineering degrees is C3 (participation in community processes that promotes sustainability), which is present in just over 23%. That is to say, almost 77% of the cells in the engineering sustainability map are not developed in the ten degrees analyzed. On the other hand, the competency with the most presence is C4 (application of ethical principles related to the values of sustainability in personal and professional behavior), with a presence of 77%. Competencies C1 (critical contextualization of knowledge by establishing interrelations with social, economic, environmental, local and/or global problems) and C2 (sustainable

use of resources and prevention of negative on the natural and social environment) are present at 55% and 53%, respectively. If we assume that the weight of each of the four competencies on the learning of sustainability is 25%, we can conclude that the presence of sustainability in the ten curriculums analyzed is 52%. Therefore, practically half of the cells of the ten engineering sustainability maps are not developed in the studied degrees.

When the sustainability presence was analyzed in each of the domain levels of the taxonomy, it was observed that, in general, the domain level "know" is the least developed in each competency, with the exception of the competency C3 (participation in community processes that promote sustainability), in which the domain level "know" is the level that develops the most. This could be because teachers believe that they should not develop the domain level "know," since it should have been developed in earlier stages of learning, and they thus focus on developing the higher levels of the taxonomy.

When the domain levels were analyzed to check how many degrees develop each level of each competency unit, it was observed that only 12.5% of the engineering sustainability map cells are developed by eight or more degrees. These cells correspond to the "demonstrate and do" level of the competency units C1.H.1. (historical perspective) and C2.EV.1. (environmental impact), and the level "know-how" of the competency unit C4.H.1. (behaves according to deontological principles). On the other hand, almost 33% of the cells of the engineering sustainability map are developed by four or less degrees. Half of these cells correspond to the "know" level of some competency unit. The competency units C2.EC.1. (economic management of an engineering project) and C3.H.1. (sustainability improved by collaborative work) are developed by four or less degrees in all their domain levels. These competency units are therefore the least present in general in the degrees under analysis.

Regarding the analysis of degrees and universities, the data show that there does not seem to be a general strategy in universities to develop sustainability in all their degrees. However, the data are compatible with the fact that some degrees do have their own strategy. This is suggested by the data in Figure 7, which show a certain correlation between the number of subjects that develop sustainability in a degree and their level of presence.

As an institution devoted to education and research, the university has a very important role to play in society. One of its salient functions is to form critical, committed, reflective and proactive citizens capable of contributing to a social transformation in accordance with the principles of sustainable development. In Europe, the European Higher Education Area provides an opportunity for change that can assist in the progress towards a more critical, committed and reflective education in the engineering degrees.

The results presented in this research are only the first step in an unprecedented study that highlights the unequal implementation of sustainability in engineering degrees. The study justifies the need for deeper future research that goes beyond a purely documentary analysis. It is necessary to analyze the degree of knowledge of sustainability acquired by students on completion of their studies and compare it with the data presented in this work. Such an analysis will enable qualitative information about the learning of sustainability in the engineering degrees to be obtained. It is also necessary to study teachers' level of knowledge of sustainability. From this information, it will be possible to design training programs for teachers with the aim of including activities related to sustainability in the subjects they teach. Furthermore, training the managers of the universities and degrees so that they capable of implementing a strategy aimed at developing sustainability in each degree is also essential. These objectives are part of the EDINSOST project, the framework for the research presented in this paper, and will therefore be pursued in future work.

**Author Contributions:** Conceptualization, F.S.-C.; data curation, F.S.-C.; formal analysis, F.S.-C., F.M.M.-P. and B.S.; funding acquisition, F.S.-C.; investigation, F.S.-C., F.M.M.-P., B.S., M.A. and I.G.; methodology, F.S.-C., F.M.M.-P., B.S., M.A. and I.G.; supervision, F.S.-C., validation, F.S.-C., F.M.M.-P., and B.S.; visualization, F.S.-C., F.M.M.-P., and B.S.; writing—original draft, F.S.-C., F.M.M.-P., B.S., M.A. and I.G.; writing—review and editing, F.S.-C.

**Funding:** This research was funded by Ministerio de Economía y Competitividad, Gobierno de España, grant number EDU2015-65574-R and by Ministerio de Ciencia, Innovación y Universidades, Gobierno de España, grant number RTI2018-094982-B-I00.

**Acknowledgments:** We want to thank the rest of the EDINSOST team for their collaboration, especially Antonio Gomera, José Manuel Muñoz, Jorge Ruíz, Rocío Valderrama and Rafael Miñano.

**Conflicts of Interest:** The authors declare no conflict of interest.

## Glossary

| | |
|---|---|
| UCO | University of Córdoba |
| UPC | Universitat Politècnica de Catalunya-BarcelonaTech |
| UPM | Universidad Politécnica de Madrid |
| BDEE | Bachelor Degree in Electrical Engineering |
| BDIE | Bachelor Degree in Informatics Engineering |
| BDME | Bachelor Degree in Mechanical Engineering |
| BDDE | Bachelor Degree in Design Engineering |
| BDCHE | Bachelor Degree in Chemical Engineering |
| BDITE | Bachelor Degree in Industrial-Technologies Engineering |

## Appendix A

Table A1 shows the complete engineering sustainability map, as presented in [25]. Column C indicates the four competencies considered:

- C1: Critical contextualization of knowledge by establishing interrelations with social, economic, environmental, local and/or global problems.
- C2: Sustainable use of resources and prevention of negative in the natural and social environment.
- C3: Participation in community processes that promotes sustainability.
- C4: Application of ethical principles related to the values of sustainability in personal and professional behaviour.

Column D indicates the dimension of sustainability in which the competency unit is framed:

- H: Holistic
- EV: Environmental
- S: Social
- EC: Economic

The rest of the columns are self-explanatory in the table.

**Table A1.** Engineering sustainability map, as presented in [25].

| | | | | | |
|---|---|---|---|---|---|
| **Engineering Sustainability Map** | | | | | |
| **C** | **D** | **Competency unit** | **Domain levels (according to simplified Miller Pyramid )** | | |
| | | | **1. KNOW** | **2. KNOW HOW** | **3. DEMONSTRATE and DO** |
| C1 | H | Has a historical perspective (state of the art) and understands social, economic and environmental problems, both locally and globally. | Knows the main causes, consequences and solutions proposed in the literature regarding the social, economic and/or environmental problems, both locally and globally. | Analyzes the different dimensions of sustainability when solving a specific problem related to engineering. | Identifies the main causes and consequences of a problem related to the sustainability that a product or a service related to engineering can have, and is able to relate them to known problems and solutions previously applied. |
| | | Is creative and innovative. Is able to see the opportunities offered by engineering to contribute to the development of more sustainable products and processes. | Has sufficient knowledge of the concepts of creativity and innovation, and about strategies to develop them. | Reflects on new ways of doing things. Knows how to use techniques that stimulate creativity, the generation of ideas, and manages them in such a way that they become an innovation. Participates actively when used. | Brings new ideas and solutions to a project related to engineering to make it more sustainable, so as to improve the sustainability of products, processes or services. |

**Table A1.** *Cont.*

| | | | Engineering Sustainability Map | | |
|---|---|---|---|---|---|
| C | D | Competency unit | Domain levels (according to simplified Miller Pyramid ) | | |
| | | | 1. KNOW | 2. KNOW HOW | 3. DEMONSTRATE and DO |
| C2 | H | Takes into account sustainability in his/her work as an engineer. | Knows the concept of cost of use, direct and indirect, of the products and services of the technologies related to engineering. Knows the strategic role that the technologies related with engineering play in the sustainability of the planet. Knows the concepts of social justice, resource reuse and circular economy. Knows the concept of social economy, the advantages of solidarity, teamwork and cooperation versus competition. Knows the principles of the economy for the common good. | Is capable of assessing the impact (positive and negative) that different products and services related to engineering have in society and in the sustainability of the planet. Knows how to assess the economic viability of a project of the engineering and whether it is compatible with the environmental and social aspects of sustainability. | Is capable of proposing sustainable projects related to engineering taking into account, holistically, the environmental, economic and social aspects. |
| | EV | Takes into account the environmental impact of his/her work as an engineer. | Knows technologies of reuse, reduction, recycling and minimization of the natural resources and residues related to a project of engineering. Knows the life cycle of the products related to engineering (construction, use and destruction/dismantling) and the concept of ecological footprint. Knows models for ecological footprint calculation. Knows metrics to measure the environmental impact of a project (e.g., pollutant emissions, resource consumption). | Is aware that products and services related to engineering have an environmental impact throughout its life. Is capable of measuring the environmental impact of the use of technologies related to engineering using appropriate metrics (e.g., pollutant emissions, and resource consumption,.). | Takes into account the environmental effects of the products and services related to engineering in the projects and technological solutions in which he / she participates. Includes in his/her projects indicators to estimate/measure these effects from the resources used by the project (e.g. energy consumption, pollutant emissions, and consumption of resources). Calculates the ecological footprint of an engineering project. |
| | S | Takes into account the social impact of his/her work as an engineer. | Knows the problems associated with accessibility, ergonomics and safety of products and projects of engineering. Knows the problems associated with social justice, equity, diversity and transparency (gender perspective, needs of the most vulnerable groups, strategies against corruption, etc.). Knows the direct and indirect consequences that the products and services related to engineering have on the society. | Knows how to assess the degree of accessibility, ergonomic quality, the level of safety and the impact on society of a product or service related to engineering. Takes into account the rights of people in their work as an engineer. Understands the need to introduce social justice, equity, diversity, transparency (gender perspective, needs of the most vulnerable groups, anti-corruption, etc.) in projects of engineering. Can assess whether an engineering project contributes to improving the common good of society. | Takes into account the aspects of accessibility, ergonomics and security in technological solutions. Takes into account social justice, equity, diversity and transparency (gender perspective, needs of vulnerable groups, combating inequality and corruption, etc.) in his/her projects. Includes in his/her projects indicators to estimate/measure how they improve the common good of society. Is able to maximize the positive impact of his/her professional activity on society. Is capable of designing engineering projects that contribute to improve the common good of society. |
| | EC | Is capable of successfully carrying out the economic management of an engineering project. | Knows basic concepts about organizations. Knows the fundamental points of a business plan. Knows the process of managing a project. Knows project-planning techniques. | Understands the different economic parts of a project: amortizations, fixed costs, variable costs, etc. Analyzes real planning cases and project budgets. | Is able to plan an engineering project (both short and long term) and to prepare a complete budget based on the material and human resources required. Is able to follow economic development of a project and detect deviations from the initial planning. Is capable of carrying out the economic management of an engineering project throughout its useful life. |

**Table A1.** *Cont.*

| | | | Engineering Sustainability Map | | |
|---|---|---|---|---|---|
| | | | Domain levels (according to simplified Miller Pyramid ) | | |
| **C** | **D** | **Competency unit** | **1. KNOW** | **2. KNOW HOW** | **3. DEMONSTRATE and DO** |
| C3 | H | Identifies when the sustainability of a project can be improved if it is done through community collaborative work. Responsibly performs collaborative work related to sustainability. | Knows the concept of community collaborative work and its implications in the transformation of society. Knows examples of projects that have been successfully implemented with community collaborative work in the field of engineering. Knows the tools of collaborative work in the field of engineering. | Given a project in the field of engineering, that includes a collaborative community work, is able to assess the implications of such work in the sustainability of the project. | Knows how to use collaborative work tools related to engineering projects. |
| C4 | H | Behaves according to the deontological principles related to sustainability. | Knows the deontological principles related to sustainability. He/she is aware that there are laws and regulations related to sustainability in his/her professional field. Knows the concept of social and corporate responsibility in general, and its possibilities and limitations. | Is able to assess the implications of the deontological principles related to sustainability in a project in the field of engineering. | Does not make decisions that contradict the deontological principles related to sustainability. Is capable of proposing solutions and strategies to promote projects in the field of engineering, consistent with these principles. |

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
