# Peer review of "A Methodology to Analyze the Presence of Sustainability in Engineering Curricula. Case of Study: Ten Spanish Engineering Degree Curricula"

_sustainability, doi:10.3390/su11174553_

Round 1
Reviewer 1 Report
The current format of the manuscript is almost well-done. Only minor revisions were needed.
The authors need to report the validity and reliability of the research measurement (map). More discussion about engineering education is needed in the literature part.
Author Response
Reviewer comments:
The current format of the manuscript is almost well-done. Only minor revisions were needed.
The authors need to report the validity and reliability of the research measurement (map).
Author’s answer:
The validity and reliability of the sustainability competency map are detailed in reference 25, so we have not considered it necessary to include them in this paper, since the map is used here as a tool (the objective of the paper is not to deepen the sustainability competency map, but use it as a tool). However, so that there is no doubt about the tool, we have included the following sentence in the text of the paper:
"The validity and reliability of the Sustainability Map is analyzed in [25]."
Reviewer comments:
More discussion about engineering education is needed in the literature part.
Author’s answer:
We have added some paragraphs in the introduction to improve the discussion on engineering education.
Reviewer 2 Report
This article proposes a method to analyze the presence of sustainability in engineering courses. The Spanish engineering degree curricula are evaluated as case studies.
Page 3: the authors mentioned that the tuning project considers five competencies (C6, C17, C23, C25, C28) related to sustainability. Moreover, Table 1 shows C1 to C4, which four competencies are analyzed from the holistic point of view whenever possible.
The authors should integrate the mentioned above 9 competencies. The related discussions and analysis are focused on C1 to C5, how about C6, C17, C23, C25, C28?
Misc.
I suggest the paper title should be considered to rename. The current title is too long and unclear. Maybe, subtitle can be used.
Author Response
Reviewer comments:
This article proposes a method to analyze the presence of sustainability in engineering courses. The Spanish engineering degree curricula are evaluated as case studies.
Page 3: the authors mentioned that the tuning project considers five competencies (C6, C17, C23, C25, C28) related to sustainability. Moreover, Table 1 shows C1 to C4, which four competencies are analyzed from the holistic point of view whenever possible.
The authors should integrate the mentioned above 9 competencies. The related discussions and analysis are focused on C1 to C5, how about C6, C17, C23, C25, C28?
Author’s answer:
The reviewer is right that this topic is not well explained. We have included the following text in the paper to clarify this point.
“The competencies defined by the CRUE-Sustainability Sectorial Commission were defined after an exhaustive review of the literature. The objective of this review was to achieve a reduced and complete set of competencies that integrated most of the sustainability competencies identified in different international studies. As can be seen, the four competencies defined by the CRUE-Sustainability Sectorial Commission have a more generic character than the five competencies defined by the Tuning project. Thus, the CRUE C4 competency contains the Tuning C6, C17, C23, and C25 competencies, while the CRUE C2 competency contains the C28 Tuning competency.”
Reviewer comments:
Misc.
I suggest the paper title should be considered to rename. The current title is too long and unclear. Maybe, subtitle can be used.
Author’s answer:
We agree that the title is too long. We were not aware of the possibility of using a subtitle. We did not find the "subtitle" style in the journal template style list. If subtitles could be used, the title would be: "A Methodology to analyze the presence of sustainability in engineering curricula", and the subtitle: "Case of study: ten Spanish Engineering degree curricula". We have put the subtitle with letter size 16, since that of the title is size 18.
Reviewer 3 Report
REVIEW REPORT
Brief Summary:
This paper is interesting. It discusses education on the world’s recent issue; sustainable development. This analyzes how sustainability appears in the curriculum of engineering education. The data included as well as the methodology are presented comprehensively.
Comments:
1. The authors are suggested to add more information in the Introduction section regarding the selection of proposed methodology compared with previous studies, such as a quantifying the integration of sustainability in energy degree programs (DOI: 10.1016/j.jclepro.2014.10.012).
2. In order to improve clarity of the method so that increasing the reproducibility by the other researchers, the authors are requested to provide more information on how reviewing in depth the learning guides of the degree subjects as written in page 6 line 216-217.
3. Regarding improving the clarity of the method, some questions may also ameliorate the clarity, such as how to check the learning guides of all degree subjects, are the guides available online, and whether the learning guides of degree subjects are similar with syllabus in some countries’ education system.
4. The authors need to clarify whether “Different subjects” in the last row of Table 2 is more appropriate compared with “Number of subjects” according to the explanation in the paragraph at page 8 line 297-302. If so, please improve the explanation of the aforementioned paragraph.
5. The section 3.5 on limitations and future work(s) are the important part of this manuscript. It becomes the consideration when other researchers apply the similar method on their curricula. However, it is still lack of further explanation especially in the first out of three limitations mentioned in the section. Meanwhile, the future work mentioned in the section only presents the work which has been doing by the author(s). It does not explain what the authors suggestions in order to perfect the study presented in the manuscript. The authors suggestion here may inspire the other researchers to improve the work presented in the manuscript while implementing the method on their education system. Such suggestions may increase the citation of this paper too.
6. The authors are suggested to revise title of the Appendix section from Appendix A to Appendix since there is only one appendix within this manuscript.
Author Response
Reviewer comments:
The authors are suggested to add more information in the Introduction section regarding the selection of proposed methodology compared with previous studies, such as a quantifying the integration of sustainability in energy degree programs (DOI: 10.1016/j.jclepro.2014.10.012).
Author’s answer:
We have added some paragraphs in the introduction to improve the discussion on engineering education, and we have included the references suggested by the reviewer, as well as the reasons why we propose the use of our methodology instead of those proposed by other studies. We have added the following text:
“The competencies defined by the CRUE-Sustainability Sectorial Commission were defined after an exhaustive review of the literature. The objective of this review was to achieve a reduced and complete set of competencies that integrated most of the sustainability competencies identified in different international studies. As can be seen, the four competencies defined by the CRUE-Sustainability Sectorial Commission have a more generic character than the five competencies defined by the Tuning project. Thus, the CRUE C4 competency contains the Tuning C6, C17, C23, C25 competencies, and the CRUE C2 competency contains the C28 Tuning competency.
The objective of this work is to present a methodology for measuring the level of sustainability presence in an engineering curriculum. Previous studies have presented proposals with similar objectives, but they consist of proposals that are less generalist. For example, in [13] a methodology is proposed to calculate the relevance ratio index of a curriculum, defined as the relative weight of renewable energy and sustainability topics for energy studies. This methodology is focused on its use in Energy Degree programs, and some changes are required for its application to other different curricula. The methodology proposed in this work, however, is general and can be used in any engineering degree. In addition, the proposed methodology can be easily applied to other non-engineering degrees simply by changing the sustainability map, as shown in (34).”
Reviewer comments:
In order to improve clarity of the method so that increasing the reproducibility by the other researchers, the authors are requested to provide more information on how reviewing in depth the learning guides of the degree subjects as written in page 6 line 216-217.
Author’s answer:
We have included more information in this regard, as proposed by the reviewer. Specifically, the following paragraph has been added:
“The assigned competencies, objectives and contents, as well as the activities carried out by the students, have been reviewed for each subject. All these aspects have been compared with the Learning Outcomes of the Engineering Sustainability Map in order to detect any coincidences. It is not enough to search for keywords within the learning guides, since in many cases the teachers who write the guides are not experts in sustainability and do not use a precise technical vocabulary. Learning guides should therefore be analyzed by sustainability experts who know the Engineering Sustainability Map well. In this work, the review of the learning guides has been carried out by members of the EDINSOST project.”
Reviewer comments:
Regarding improving the clarity of the method, some questions may also ameliorate the clarity, such as how to check the learning guides of all degree subjects, are the guides available online, and whether the learning guides of degree subjects are similar with syllabus in some countries’ education system.
Author’s answer:
We think that this issue has been answered in the previous point. In the case of the Spanish university system, all subject learning guides are available online through the school's website or through the subject's website. However, Spanish degrees must be previously accredited before they begin to be taught. To obtain this accreditation, the Degrees must present a document called "Verifica" to the national accreditation agency (ANECA). This document includes the competencies of the Degree, its distribution among the subjects, and a definition of the objectives, methodology, contents and activities of each subject. This document has also been consulted in this work, although it is usually less detailed and less updated than the subject learning guides. We have not mentioned this aspect in the paper because we understand that "Verifica" is a particular case of the Spanish university system.
Reviewer comments:
The authors need to clarify whether “Different subjects” in the last row of Table 2 is more appropriate compared with “Number of subjects” according to the explanation in the paragraph at page 8 line 297-302. If so, please improve the explanation of the aforementioned paragraph.
Author’s answer:
We have included in the text the following explanation:
“For example, a subject X of the UPC BDEE could develop domain levels L1, L2 and L3 of the Competency Unit C1.H.1. This subject should be considered only once for counting the number of subjects that develop sustainability in the Degree, but if the row "Different subjects" were the result of the sum of the corresponding columns, the subject X would be counted three times in this case. “
For this reason, we consider the denomination "different subjects" more appropriate than "number of subjects".
Reviewer comments:
The section 3.5 on limitations and future work(s) are the important part of this manuscript. It becomes the consideration when other researchers apply the similar method on their curricula. However, it is still lack of further explanation especially in the first out of three limitations mentioned in the section. Meanwhile, the future work mentioned in the section only presents the work which has been doing by the author(s). It does not explain what the authors suggestions in order to perfect the study presented in the manuscript. The authors suggestion here may inspire the other researchers to improve the work presented in the manuscript while implementing the method on their education system. Such suggestions may increase the citation of this paper too.
Author’s answer:
We agree to the reviewer. We have added the following paragraph at the end of first limitation:
“Spain has 87 registered universities and 231 Engineering Degrees (some of them with very similar names), so the three universities and six degrees analyzed in this work constitute a very small sample. However, the three universities considered belong to the group of universities with a greater number of students and a higher prestige within the Spanish university system. Therefore, although the results presented in this paper cannot be generalized, the authors consider them to be a representative sample of the presence of sustainability in Spanish Engineering Degrees. The complicity (and willingness) of a large number of teachers from most Spanish universities would be necessary in order to carry out a study with any claim to statistical validity. In addition, these teachers should be trained to analyze the learning guides of the subjects and relate their content to the Learning Outcomes of the Engineering Sustainability Map. We believe that it is very difficult for these conditions to be met in the short term, so it is important to have studies of this nature which, while they may not ensure statistical validity, would allow us to acquire a vision of the presence of sustainability in some Engineering Degrees. The objective of the EDINSOST2 project (2019-2022), a continuation of the EDINSOST project (2016-2019), is to improve the sustainability training of teachers of the Spanish university system, thereby enabling them to use the tools designed by EDINSOST in their degrees.”
Morover, we have included the following paragraph at the end of the Section:
“The authors consider that it is necessary to advance in this direction by measuring not only the presence of sustainability in the curricula, but also the level of graduate training. This information is essential to determine how sustainability is being developed in the different university degrees. From this diagnosis, curriculum designers can identify the weaknesses of current curricula and include Learning Outcomes that are not developed or not developed enough. Such a diagnosis would also reveal the areas in which the teachers need more training, and also enable the design of a training program aimed at covering the needs of each degree. The Engineering Sustainability Map provides the perfect tool for guiding these objectives.”
Reviewer comments:
The authors are suggested to revise title of the Appendix section from Appendix A to Appendix since there is only one appendix within this manuscript.
Author’s answer:
Done.
Round 2
Reviewer 2 Report
The authors have great efforts for enhancing and improving their first submitted draft. The authors have satisfyingly replied to my all questions. So, I think that the revised manuscript can be accepted for publication.